# A Modified Keystone-Based Forward-Looking Arc Array Synthetic Aperture Radar 3D Imaging Method

**DOI:** 10.3390/s23052674

**Published:** 2023-02-28

**Authors:** Xiaofan Zhu, Pingping Huang, Wei Xu, Weixian Tan, Yaolong Qi

**Affiliations:** 1College of Information Engineering, Inner Mongolia University of Technology, Hohhot 010051, China; 2Inner Mongolia Key Laboratory of Radar Technology and Application, Hohhot 010051, China

**Keywords:** synthetic aperture radar (SAR), arc array, forward motion of the carrier platform, first-order term approximation method, improved keystone-based processing, spatial resolution

## Abstract

An arc array synthetic aperture radar (AA-SAR) is a new type of omnidirectional observation and imaging system. Based on linear array 3D imaging, this paper introduces a keystone algorithm combined with the arc array SAR 2D imaging method and proposes a modified 3D imaging algorithm based on keystone transformation. The first step is to discuss the target azimuth angle, retain the far-field approximation method of the first-order term, analyze the influence of the forward motion of the platform on the along-track position, and realize the two-dimensional focusing of the target slant range–azimuth direction. The second step is to redefine a new azimuth angle variable in the slant-range along-track imaging and use the keystone-based processing algorithm in the range frequency domain to eliminate the coupling term generated by the array angle and the slant-range time. The corrected data are used to perform along-track pulse compression to obtain the focused image of the target and realize the three-dimensional imaging of the target. Finally, in this article, the spatial resolution of the AA-SAR system in the forward-looking state is analyzed in detail, and the change in the spatial resolution of the system and the effectiveness of the algorithm are verified through simulation.

## 1. Introduction

Synthetic aperture radar has become one of the most attractive remote sensing detection tools because of its all-day working ability, high resolution, and large viewing angle [1,2,3,4]. Conventional synthetic aperture radars can reconstruct two-dimensional images of the detection area [5,6,7] with all-day operation capability, while avoiding adverse climate effects. In recent years, an arc array SAR imaging system device/method that breaks through the single observation angle of the linear array SAR system and is no longer limited to front-looking, down-looking, or side-looking imaging has been proposed [8,9,10,11]. The existing arc array two-dimensional imaging system is receiving more and more attention. For example, an arc array bistatic SAR with a stationary transmitter has been proposed by relevant scholars. AA-BiSAR has the advantages of both BiSAR and AA-SAR, which can obtain highly efficient data acquisition and reduce the vulnerability of arc array monostatic SAR. In AA-BiSAR, high-range resolution can be achieved by the high-power wide-bandwidth linear frequency modulation signal transmitted by the stationary transmitter far from the passive receiver. In the azimuth direction, multiple antenna array elements are arranged along a cylinder to form an arc-synthetic aperture, and the switched array antenna channels scan quickly over the circular aperture. Based on this, arc array bistatic SAR with a moving transmitter is proposed; it has the advantages of good concealment and the ability to expand the imaging scene, as well as improved flexibility of the system. AA-SAR with one moving transmitter can be applied to improve the ability of a helicopter to perceive and detect the surrounding targets; since the transmitter can move, multiple low-cost receivers can share the same transmitter to reduce the system cost. Because the arc array SAR has a special observation geometric model and azimuth equal-angle sampling method, the traditional imaging algorithm used for the linear array SAR is no longer suitable for the arc array SAR. The imaging processing method of an arc array SAR is necessary. However, in many application scenarios, two-dimensional images cannot meet the requirements. In the existing literature, there is very little research on the 3D imaging of arc arrays. Therefore, how to use the existing system to achieve 3D imaging in low-altitude airspace becomes an urgent problem to be solved.

SAR 3D imaging technology is an extension of SAR 2D imaging, which solves problems such as the overlapping and inversion of traditional imaging. It can realize the reconstruction of the spatial position of the target in the observation area and has great application potential in the fields of battlefield reconnaissance, environmental monitoring, terrain mapping, etc. It has become one of the research hotspots in the field of radar imaging. In recent years, researchers at home and abroad have actively explored SAR three-dimensional imaging technology and achieved results in developing imaging models, imaging theory, imaging algorithms, and experimental systems. At present, common three-dimensional imaging technologies can be divided into the following types according to different imaging modes: InSAR and ROSAR. Assuming that only one scatterer in each resolution unit generates echo signals, InSAR 3D imaging obtains the elevation information of the target through the inversion of the phase difference information [12,13,14]. Multi-baseline SAR tomography is realized through multiple altitude baselines [15,16,17]. The altitude baseline can be realized by time-sharing multiple passes or multiple antennas at the same time, but the altitude blur is mainly due to the unavoidable problem of uneven track distribution. The forward-looking 3D imaging of the linear array antenna can realize a certain resolution imaging of the sector area directly in front of the aircraft route [18,19,20]. However, owing to the limitations of aerodynamics and mechanical principles, the array antenna generally cannot be made very large, resulting in poor azimuthal resolution. In ROSAR 3D technology [21,22], the antenna moves in a uniform circular motion along with the wings, while the carrier platform moves forward to form a horizontal composite surface in the air, combined with the wideband signal emitted in the forward direction. ROSAR uses three-dimensional cognition to detect the scene ahead. However, with the large antenna error with mechanical rotation, it is impossible to obtain accurate equivalent sampling point information, which brings great challenges to later imaging processing. The arc array synthetic aperture radar forward-looking 3D imaging technology works in a new imaging mode, which breaks through the shortcomings of the above four geometric configurations and can meet the requirements of specific aircraft platforms for the 3D imaging of forward targets.

This paper introduces an accurate and efficient 3D imaging mode for arc array SAR—the forward mode. In the imaging algorithm proposed in this paper, the keystone algorithm is introduced based on the three-dimensional imaging of the linear array combined with the two-dimensional imaging method of the arc array to modify the algorithm so that the arc array SAR has three-dimensional imaging capabilities, making the algorithm suitable for arc array. This mode can ensure the safety of the forward flight of the carrier platform and realize the three-dimensional imaging of the scene ahead. This paper simplifies the echo signal and divides the three-dimensional imaging process of the low-altitude ground-mounted reconnaissance platform into two processing steps: slant range–azimuth and slant range–along track. However, this processing introduces two problems: On the one hand, the existence of the target azimuth makes the trigonometric function always exist in the target slant-range equation, which is inconvenient for subsequent processing. On the other hand, the range-cell offset is a common problem in various high-resolution SAR systems, and the imaging capability is highly affected by distance migration correction. In view of the above two problems, first, this paper discusses in detail the target azimuth angle, analyzes the influence of the forward motion of the platform on the along-track direction, and uses approximate processing to make subsequent imaging processing more convenient. Second, the trapezoidal transformation is introduced in this paper, correcting the range migration caused by the coupling term between the slant range and azimuth. Keystone transformation is a data scale remapping method [23,24,25], usually used for range offset correction of objects. Because the transformation involves azimuth and trigonometric functions, the uniqueness of the nonlinear transformation is also explained and demonstrated in this paper. In addition, in this mode, the carrier platform moves forward while the arc array antenna adopts the electronic scanning mode. The opening or closing of the antenna element corresponding to the direction of the target position is selected and controlled by the high-speed microwave switch network, thereby forming a horizontal synthetic front in the air, combined with the broadband signal emitted in the oblique direction, so that it can provide three-dimensional imaging of the scene ahead. It does not increase the hardware complexity of the original system but solves the huge challenge caused by the serious coupling of echo data in the three directions of the slant range–azimuth track.

This paper is organized as follows. In Section 2, the imaging geometric model of the arc array synthetic aperture radar and the observation geometric model of the aircraft platform in the low-altitude reconnaissance mode are given. To facilitate subsequent analysis, the corresponding variable transformation between the coordinate systems is given. At the same time, the echo signal model based on imaging geometry is established. Section 3 analyzes the azimuth angle of the target that is not on the track axis and proposes an imaging algorithm based on KT transform. In Section 4, the spatial resolution of arc array synthetic aperture radar in forward mode is analyzed, and the relationship between azimuth and azimuth is given. In Section 5, numerical simulation experiments are carried out on point targets, and the imaging results of the targets verify the proposed imaging method. Finally, conclusions are given in Section 6.

## 2. Arc Array Synthetic Aperture Radar (AA-SAR)

### 2.1. Imaging Geometry

As a new type of airborne SAR observation system, arc array SAR adopts a wide-area circular-beam scanning method, which has a larger observation area and accelerates the speed of radar acquisition of scene information. Among them, the arc array antenna is the most important part of the system, which is related to whether the whole imaging system can realize all-around fast imaging. In the design of the SAR system, the frequency-modulated continuous-wave (FMCW) transceiver is used, the transmitting antenna and the receiving antenna are arranged separately, and the FMCW system is small in size, light in weight, simple in structure, and easy to use. Compared with pulsed radar, the implementation is easy, costs are low, and the burden of the carrier platform is reduced. In terms of the antenna configuration and imaging geometry of the arc array, its sampling model differs from that of the traditional linear array SAR. Linear arrays sample at equal intervals in azimuth, while arc arrays sample at equal angles in azimuth. The sampling models of the two imaging modes are different, so traditional linear array imaging algorithms cannot be directly used for the imaging processing of arc array SAR; thus, the imaging processing method for arc array SAR needs to be studied.

Figure 1a shows the structural diagram of the arc antenna. The transmitting array antenna and receiving array antenna are evenly arranged along the arc-angle direction, which is called the array direction here. The angular spacing between adjacent antenna elements is set to the same $\Delta \varphi_{\mathrm{Interval}}$, the radius is $R_{\mathrm{arc}}$, $\varphi_{0}$ represents the synthetic aperture angle of the arc array antenna, and the azimuth beamwidth of the antenna element is expressed as β. At the same time, $\varphi_{0}$ and β jointly determine the observation range of the effective azimuth of the system. The geometric scene of arc array SAR imaging is shown in Figure 1b. The range resolution is realized by transmitting the bandwidth signal. Meanwhile, the azimuth resolution is achieved through the arc synthetic aperture formed by multiple antenna array elements arranged along the circular arc.

### 2.2. Platform Observation Geometry in Forward Mode

Figure 2 shows the geometric model of the scene imaging in the forward mode. The forward direction of the helicopter is defined along the X-axis, the intersection direction is defined along the Y-axis, the elevation direction is defined along the Z-axis, and the r-axis represents the slant distance. The platform is set to move forward in a straight line at a constant speed v, where η is the time variable along the track, the pitch angle is θ, the pitch beam width is ε, and the antenna azimuth beam width is β. The origin of the observation scene coordinates is O, and the center point of the arc array antenna element is $O^{\prime }$. The cylindrical coordinates of the equivalent sampling point $P_{\mathrm{apc}}$ of the arc array are expressed as $P_{\mathrm{apc}}\left(\varphi ,R_{\mathrm{arc}},h_{\mathrm{arc}}\right)$, the angle of the horizontal plane of the adjacent equivalent sampling point $P_{\mathrm{apc}}$ is expressed as φ, the arc array antenna radius is denoted as $R_{\mathrm{arc}}$, and the platform height is denoted as $h_{\mathrm{arc}}$. The position coordinate of the imaging target $P_{n}$ is expressed as $P_{n}(\varphi_{n},R_{n},h_{n})$, its azimuth angle is expressed as $\varphi_{n}$, the ground distance is expressed as $R_{n}$, and the height is expressed as $h_{n}$. When the platform transmits signals, the helicopter flies forward, and $x_{l}$ indicates that $O^{\prime }$ moves to a specific position on the X-axis. We converted the data stored in the cylindrical coordinate system to the rectangular coordinate system according to the following formula for the analysis and discussion below.
(1)$$\left\{\begin{array}{l} x_{\mathrm{apc}}=R_{\mathrm{arc}}\cos\varphi +x_{l},y_{\mathrm{apc}}=R_{\mathrm{arc}}sin\varphi ,z_{\mathrm{apc}}=h_{\mathrm{arc}}\\ x_{0}=R_{n}\cos\varphi_{n},y_{0}=R_{n}\sin\varphi_{n},z_{0}=h_{n} \end{array}\right.$$

## 3. Analysis of Change in the Azimuth Angle

### 3.1. Echo Signal Model

The arc array SAR imaging adopts the frequency-modulated continuous-wave (FMCW) system, and the transmitted signal is expressed as
(2)$$\begin{array}{l} s_{t}=\mathrm{rect}(\frac{\tau }{T_{p}})\exp(j2\pi f_{0}\tau +\pi K_{r}\tau^{2})\\ \mathrm{rect}(\frac{\tau }{T_{p}})=\left\{\begin{array}{l} 1,\tau \in \left[-T_{p}/2,T_{p}/2,\right]\\ 0,\mathrm{else} \end{array}\right. \end{array}$$
where $f_{0}$ refers to the carrier frequency, $K_{r}$ is the frequency modulation rate, τ is the distance-to-fast time variable, and $T_{p}$ is the pulse width of the transmitted signal. The echo signal scattered by the imaging target is received by the equivalent sampling point and expressed as
(3)$$\begin{array}{l} ss_{\mathrm{re}}(\tau ;\varphi ;R_{m})=\sigma_{n}(\varphi_{n};R_{n};h_{n})\mathrm{rect}\left[\frac{\tau -\frac{2R_{m}}{c}}{T_{p}}\right]\\ \;\cdot \mathrm{rect}\left[\frac{\varphi -\varphi_{n}}{\varphi_{s}}\right]\exp\left\{j\pi \left[2f_{0}\left(\tau -\frac{2R_{m}}{c}\right)+K_{r}{\left(\tau -\frac{2R_{m}}{c}\right)}^{2}\right]\right\} \end{array}$$

In Formula (3), c represents the speed of light, $\sigma_{n}(\varphi_{n};R_{n};h_{n})$ is the backscattering coefficient of the target, and $\varphi_{s}$ represents the beam width of the antenna array element along the horizontal array direction.

Mixing the echo signal with the transmitted signal, which is De-chirp processing, the resulting demodulated signal can be expressed as
(4)$$\begin{array}{c} ss_{\mathrm{IF}}(\tau ;\varphi ;R_{m})=\sigma_{n}(\varphi_{n};r_{n};h_{n})\mathrm{rect}\left(\frac{\tau -2R_{m}/c}{T_{p}}\right)\mathrm{rect}\left(\frac{\varphi -\varphi_{n}}{\varphi_{s}}\right)\\ \cdot \exp\left\{-j\frac{4\pi R_{m}\left(f_{0}+K_{r}\tau \right)}{c}\right\}\\ \;\cdot \exp\left\{-j\pi K_{r}{\left(\tau -\frac{2R_{m}}{c}\right)}^{2}\right\} \end{array}$$

The second exponential term $j\pi K_{r}{\left(\tau -2R_{m}/c\right)}^{2}$ in Equation (4) is the residual video phase term (RVP), and the De-chirp signal after compensation by the RVP term can be given by
(5)$$\begin{array}{c} ss_{\mathrm{rvp}}(\tau ;\varphi ;R_{m})=\sigma_{n}(\varphi_{n};r_{n};h_{n})\mathrm{rect}\left(\frac{\tau -2R_{m}/c}{T_{p}}\right)\mathrm{rect}\left(\frac{\varphi -\varphi_{n}}{\varphi_{s}}\right)\\ \cdot \exp\left\{-j\frac{4\pi R_{m}\left(f_{0}+K_{r}\tau \right)}{c}\right\} \end{array}$$

### 3.2. Target Azimuth Analysis

According to the azimuth section view shown in Figure 3, the instantaneous slope distance $R_{m}$ from any target in the imaging area to the equivalent sampling point can be written as
(6)$$R_{m}=\sqrt{{\left(R_{\mathrm{arc}}\cos\varphi +x_{l}-x_{0}\right)}^{2}+{\left(R_{\mathrm{arc}}sin\varphi -y_{0}\right)}^{2}+{\left(h_{\mathrm{arc}}-z_{0}\right)}^{2}}$$

When the platform center $O^{\prime }$ moves to $x_{l}$, and the carrier platform detects the target in the front area, $L=h_{\mathrm{arc}}[\tan(\theta +\varepsilon /2)-\tan(\theta -\varepsilon /2)]$ is defined as the detection range along the track, which is the synthetic aperture length along the track. Here, for the convenience of analysis, the platform moves to the center point $x_{\mathrm{mid}}$ of the detectable range L in an ideal situation, $x_{l}=x_{\mathrm{mid}}+\Delta x_{l}$ is re-expressed, and $\varphi_{\mathrm{mid}}=a\tan\left[y_{0}/\left(x_{0}-x_{\mathrm{mid}}\right)\right]$ represents the azimuth of the target at the time when the axis position is at $x_{\mathrm{mid}}$. At this time, the ground distance of the point target when the axis position is at $x_{l}$ can be expressed by
(7)$$\begin{array}{l} R_{x_{l}}=\sqrt{{\left(x_{0}-x_{l}\right)}^{2}+{y_{0}}^{2}}\\ \;=\sqrt{{\left[x_{0}-\left(x_{\mathrm{mid}}+\Delta x_{l}\right)\right]}^{2}+{y_{0}}^{2}}\\ \;=R_{\mathrm{mid}}-\cos\left(\varphi_{\mathrm{mid}}\right)\Delta x_{l}+\frac{\sin^{2}\left(\varphi_{\mathrm{mid}}\right)\Delta {x_{l}}^{2}}{2R_{\mathrm{mid}}}+o\left[{\left(\frac{\Delta x_{l}}{R_{\mathrm{mid}}}\right)}^{3}\right] \end{array}$$
where $R_{\mathrm{mid}}=\sqrt{{\left(x_{0}-x_{\mathrm{mid}}\right)}^{2}+{y_{0}}^{2}}$ represents the ground distance of the point target when the axis position is at $x_{\mathrm{mid}}$.

Define Ω as the shortest distance from the array center $O^{\prime }$ to the target. For the convenience of expression in the following text, substitute (7) into the replacement and perform a Taylor-series expansion, which is expressed as follows:
(8)$$\begin{array}{l} \Omega =\sqrt{{\left(R_{x_{l}}-R_{\mathrm{arc}}\right)}^{2}+{\left(h_{\mathrm{arc}}-z_{0}\right)}^{2}}\\ \;=\Omega_{\mathrm{mid}}-\cos\theta \cos\left(\varphi_{\mathrm{mid}}\right)\Delta x_{l}+\frac{\sin^{2}\theta }{2\Omega_{\mathrm{mid}}}\cos^{2}\left(\varphi_{\mathrm{mid}}\right)\Delta {x_{l}}^{2}\\ \;+\frac{\cos\theta \sin^{2}\theta }{2{\Omega_{\mathrm{mid}}}^{2}}\cos^{3}\left(\varphi_{\mathrm{mid}}\right)\Delta {x_{l}}^{3}+o\left[{\left(-\cos\left(\varphi_{\mathrm{mid}}\right)\Delta x_{l}\right)}^{4}\right] \end{array}$$Here, when the platform position moves to $x_{\mathrm{mid}}$, the shortest distance between $O^{\prime }$ and the target is expressed as $\Omega_{\mathrm{mid}}=\sqrt{{\left(R_{\mathrm{mid}}-R_{\mathrm{arc}}\right)}^{2}+{\left(h_{\mathrm{arc}}-z_{0}\right)}^{2}}$. At this time, the instantaneous slope distance can be rewritten as
(9)$$R_{m}=\sqrt{\Omega^{2}+{R_{\mathrm{arc}}}^{2}-2\Omega R_{\mathrm{arc}}\cos\theta \cos\left(\varphi -\varphi_{x_{l}}\right)}$$

## 4. Arc Array SAR 3D Imaging Algorithm

This section analyzes the azimuth angle of the target, which is very important for the approximate processing of the distance between the platform and the target and gives the specific imaging algorithm flow. The complete imaging flow chart of the arc array synthetic aperture radar low-flying reconnaissance mode is shown in Figure 4. After demodulating the echo signal, perform RVP compensation, and focus the processed echo signal in the azimuth and slant directions to obtain a single SAR image. Then, the coupling between the range frequency domain variable and the azimuth angle is successfully eliminated by keystone transformation, and the focused image is outputted after eliminating the two-dimensional coupling term and along-track matched filtering.

### 4.1. Azimuth–Slant Range Processing

In Equation (5), let $f=f_{0}+K_{r}\tau$; the echo signal demodulated to the baseband signal can be written as
(10)$$\begin{array}{cc} ss(\tau ;\varphi ;R_{m}) & =\sigma_{n}(\varphi_{n};r_{n};h_{n})\mathrm{rect}\left(\frac{\tau -2R_{m}/c}{T_{p}}\right)\mathrm{rect}\left(\frac{\varphi -\varphi_{n}}{\varphi_{s}}\right)\\ & \cdot \exp\left[-j4\pi f\frac{R_{m}}{c}\right] \end{array}$$

We can easily find that when the carrier platform moves forward at a certain sampling point along the heading, a synthetic aperture is obtained in the azimuth direction so that the target in the front scene can be distinguished. After the above echo signals are post-processed by focusing on the azimuth and oblique directions, the obtained single SAR image can be expressed as
(11)$$\begin{array}{cc} ss(\tau ;\varphi ;R_{m}) & =A_{0}\sin c\left[B_{r}\left(\tau -\frac{2R_{m}}{c}\right)\right]\sin c\left[\frac{L_{\mathrm{eff}}}{\lambda R_{m}}\left(\varphi -\varphi_{n}\right)\right]\\ & \cdot \exp\left[-j4\pi \frac{R_{m}}{\lambda }\right] \end{array}$$
where $A_{0}$ is the amplitude of the target signal after azimuth–slant range focusing, the signal distance to bandwidth is expressed as $B_{r}$, and $L_{\mathrm{eff}}$ is the effective synthetic aperture of the target azimuth [26,27].

By intercepting any azimuth direction $\varphi_{n}=\varphi_{x_{l}}$, the two-dimensional data expression of the slant-range direction and the along-track direction can be obtained as
(12)$$\begin{array}{c} ss(\tau ;x_{l};\varphi_{n}=\varphi_{x_{l}})=A_{1}\sin c\left[B_{r}\left(\tau -\frac{2R_{m}}{c}\right)\right]\\ \;\cdot \exp\left[-j4\pi \frac{R_{m}}{\lambda }\right] \end{array}$$
where $A_{1}=\sigma_{n}(\varphi_{n};r_{n};h_{n})\sin c\left[\frac{L_{\mathrm{eff}}}{\lambda R_{m}}\left(\varphi -\varphi_{x_{l}}\right)\right]$. We ignore it in subsequent imaging processing to simplify the process.

### 4.2. Improved Transformation Based on Keystone

Fourier transform is performed on the signal in Equation (12) along the slant range; the result can be computed as follows:(13)$$\begin{array}{c} S_{1}(f_{r};x_{l};\varphi_{n}=\varphi_{x_{l}})=A_{2}\mathrm{rect}\left(\frac{f_{r}}{2B_{r}}\right)\\ \;\cdot \exp\left[-j4\pi \frac{\left(f_{r}+f_{0}\right)R_{m}}{c}\right] \end{array}$$
where $f_{r}$ represents the range frequency domain variable. It can be obtained from the above analysis that in the actual situation $R_{\mathrm{arc}}\ll \Omega$, the Taylor expansion of the slant range is performed, and the first-order term far-field approximation method is retained, so $R_{m}$ can be approximately expressed as
(14)$$R_{m}=\Omega -R_{\mathrm{arc}}\cos\theta \cos\left(\varphi -\varphi_{x_{l}}\right)$$

Substituting the approximate slope distance into Equation (13), the signal can be written as
(15)$$\begin{array}{cc} S_{2}(f_{r};x_{l};\varphi_{n}=\varphi_{x_{l}}) & =A_{2}\mathrm{rect}\left(\frac{f_{r}}{2B_{r}}\right)\\ & \cdot \exp\left\{\begin{array}{l} -j4\pi \frac{\left(f_{r}+f_{0}\right)}{c}\\ \cdot \left[\Omega -R_{\mathrm{arc}}\cos\theta \cos\left(\varphi -\varphi_{x_{l}}\right)\right] \end{array}\right\} \end{array}$$

For the convenience of analysis, Formula (15) can be rewritten as
(16)$$\begin{array}{cc} S_{2}(f_{r};x_{l};\varphi_{n}=\varphi_{x_{l}}) & =A_{2}\mathrm{rect}\left(\frac{f_{r}}{2B_{r}}\right)\exp\left\{-j4\pi \frac{\left(f_{r}+f_{0}\right)}{c}\Omega \right\}\\ & \cdot \exp\left\{j4\pi \frac{f_{r}}{c}R_{\mathrm{arc}}\cos\theta \cos\left(\varphi -\varphi_{x_{l}}\right)\right\}\\ & \cdot \exp\left\{j4\pi \frac{f_{0}}{c}R_{\mathrm{arc}}\cos\theta \cos\left(\varphi -\varphi_{x_{l}}\right)\right\} \end{array}$$

According to the above formula, in the second exponential term $\exp\left\{j4\pi \frac{f_{r}}{c}R_{\mathrm{arc}}\cos\theta \cos\left(\varphi -\varphi_{x_{l}}\right)\right\}$, the range frequency domain variable $f_{r}$ is coupled with the azimuth angle φ. The distance cell migration can be corrected by removing the coupling before along-track focusing is performed. In this paper, the method of keystone transformation is used to write the transformation relationship between the target azimuth angle φ and the virtual sample α as Equation (17), and the distance travel term is eliminated by redefining a virtual azimuth sample α for conversion.
(17)$$\cos\left(\alpha -\varphi_{x_{l}}\right)=\frac{f_{r}}{f_{r}+f_{0}}\cos\left(\varphi -\varphi_{x_{l}}\right)$$

Since the transformation involves trigonometric functions, we next analyze the uniqueness of the result as follows. In the actual situation, because of the difference in the distance of the observation target, the relationship between target ground distance and array distribution affects the antenna synthetic aperture angle $\varphi_{0}$. In general, it is best to select the synthetic aperture angle of the array antenna to be smaller than the beam width of the antenna array, which is $\varphi_{0}\ll \beta$. However, here, we consider taking $\varphi_{0}=\beta$ in the ideal state to obtain a better resolution in the array direction because the arc array antenna selects and controls the opening or closing of the corresponding azimuth antenna element according to the target position through the high-speed microwave switch. Therefore, in the actual case $\varphi -\varphi_{x_{l}}\leq \beta /2\leq 90^{\circ }$, so in the angular range of $\left[0^{\circ },90^{\circ }\right]$, $\cos\left(\varphi -\varphi_{x_{l}}\right)$ is a monotonic function, so Formula (17) can guarantee the uniqueness after transformation. Therefore, after substituting (17) into (16) and undergoing keystone transformation, the equation can be re-expressed as Equation (18):(18)$$\begin{array}{cc} S_{3}(f_{r};x_{l};\varphi_{n}=\varphi_{x_{l}}) & =A_{2}\mathrm{rect}\left(\frac{f_{r}}{2B_{r}}\right)\\ & \cdot \exp\left\{-j4\pi \frac{\left(f_{r}+f_{0}\right)}{c}\left[\Omega -R_{\mathrm{arc}}\cos\theta \cos\left(\alpha -\varphi_{x_{l}}\right)\right]\right\}\\ & \cdot \exp\left\{j4\pi \frac{f_{0}}{c}R_{\mathrm{arc}}\cos\theta \cos\left(\varphi -\varphi_{x_{l}}\right)\right\} \end{array}$$

It can be seen that after the keystone transformation, the coupling term between the range frequency domain variable $f_{r}$ and the azimuth angle φ is successfully eliminated, and the range cell migration is corrected.

For the AA-SAR in the forward mode, its main task is to image the point target in the front or oblique front area. Within the detectable range of the radar, the scene’s width is relatively small, which means the azimuth angle of the target is small. Therefore, within an appropriate range, for the same point target, we ignore the azimuth difference between the different axis $x_{l}$ positions when the aircraft moves forward; in the following processing, it can be approximated as $\varphi_{x_{l}}=\varphi_{\mathrm{mid}}$. According to Expansion (10), the expression of Equation (18) in the along-track frequency domain can be written as
(19)$$\begin{array}{cc} S_{4}(K_{w};K_{x};\varphi_{n}=\varphi_{\mathrm{mid}}) & =A_{3}\mathrm{rect}\left(\frac{f_{r}}{2B_{r}}\right)\\ & \cdot \exp\left\{j4\pi \frac{f_{0}}{c}R_{\mathrm{arc}}\cos\theta \cos\left(\varphi -\varphi_{\mathrm{mid}}\right)\right\}\\ & \cdot \exp\left\{-j2K_{w}\left[\Omega_{\mathrm{mid}}+x_{\mathrm{mid}}\cos\theta \cos\left(\alpha -\varphi_{\mathrm{mid}}\right)\right]\right\}\\ & \cdot \exp\left\{-jK_{x}x_{\mathrm{mid}}\right\}\\ & \cdot \exp\left\{j\left[\begin{array}{l} \frac{\Omega_{\mathrm{mid}}\lambda {K_{x}}^{2}}{8\pi \sin^{2}\theta \cos^{2}\left(\alpha -\varphi_{\mathrm{mid}}\right)}\\ +\frac{\Omega_{\mathrm{mid}}\lambda^{2}\cos\theta {K_{x}}^{3}}{32\pi^{2}\sin^{4}\theta \cos^{3}\left(\alpha -\varphi_{\mathrm{mid}}\right)} \end{array}\right]\right\}\\ & \cdot \exp\left\{-j\left[\begin{array}{l} \frac{\Omega_{\mathrm{mid}}\lambda {K_{x}}^{2}}{8\pi \sin^{2}\theta \cos^{2}\left(\alpha -\varphi_{\mathrm{mid}}\right)f_{0}}\\ +\frac{\Omega_{\mathrm{mid}}\lambda^{2}\cos\theta {K_{x}}^{3}}{16\pi^{2}\sin^{4}\theta \cos^{3}\left(\alpha -\varphi_{\mathrm{mid}}\right)f_{0}} \end{array}\right]f_{r}\right\} \end{array}$$
where $K_{w}$ represents the distance wavenumber domain variable and $K_{x}$ represents the along-track wavenumber domain variable. It can be seen from the above formula that the fifth exponential term is a two-dimensional coupling term, and Equation (20) gives the decoupling function to correct the coupling term.
(20)$$H_{1}\left(K_{w};K_{x};\varphi_{n}=\varphi_{\mathrm{mid}}\right)=\exp\left\{j\left[\begin{array}{l} \frac{\Omega_{\mathrm{mid}}\lambda {K_{x}}^{2}}{8\pi \sin^{2}\theta \cos^{2}\left(\alpha -\varphi_{\mathrm{mid}}\right)f_{0}}\\ +\frac{\Omega_{\mathrm{mid}}\lambda^{2}\cos\theta {K_{x}}^{3}}{16\pi^{2}\sin^{4}\theta \cos^{3}\left(\alpha -\varphi_{\mathrm{mid}}\right)f_{0}} \end{array}\right]f_{r}\right\}$$

The signal after the correction of the two-dimensional coupling term is subjected to the inverse fast Fourier transform of the slant range, and the signal in the frequency domain along the track in the time domain of the slant range is obtained as follows:(21)$$\begin{array}{cc} S_{5}(\tau ;K_{x};\varphi_{n}=\varphi_{\mathrm{mid}}) & =A_{4}\sin c\left\{B_{r}\left[\tau -2\frac{\Omega_{\mathrm{mid}}+x_{\mathrm{mid}}\cos\theta \cos\left(\alpha -\varphi_{\mathrm{mid}}\right)}{c}\right]\right\}\\ & \cdot \exp\left\{j4\pi \frac{f_{0}}{c}R_{\mathrm{arc}}\cos\theta \cos\left(\varphi -\varphi_{\mathrm{mid}}\right)\right\}\\ & \cdot \exp\left\{-j\frac{4\pi }{\lambda }\left[\Omega_{\mathrm{mid}}+x_{\mathrm{mid}}\cos\theta \cos\left(\alpha -\varphi_{\mathrm{mid}}\right)\right]\right\}\\ & \cdot \exp\left\{-jK_{x}x_{\mathrm{mid}}\right\}\\ & \cdot \exp\left\{j\left[\begin{array}{l} \frac{\Omega_{\mathrm{mid}}\lambda {K_{x}}^{2}}{8\pi \sin^{2}\theta \cos^{2}\left(\alpha -\varphi_{\mathrm{mid}}\right)}\\ +\frac{\Omega_{\mathrm{mid}}\lambda^{2}\cos\theta {K_{x}}^{3}}{32\pi^{2}\sin^{4}\theta \cos^{3}\left(\alpha -\varphi_{\mathrm{mid}}\right)} \end{array}\right]\right\} \end{array}$$

The fourth exponential term in Equation (21) is the along-track phase modulation information, and the along-track matching function is constructed as
(22)$$H_{2}(\tau ;K_{x};\varphi_{n}=\varphi_{\mathrm{mid}})=\exp\left\{-j\left[\frac{\Omega_{\mathrm{mid}}\lambda {K_{x}}^{2}}{8\pi \sin^{2}\theta \cos^{2}\left(\alpha -\varphi_{\mathrm{mid}}\right)}+\frac{\Omega_{\mathrm{mid}}\lambda^{2}\cos\theta {K_{x}}^{3}}{32\pi^{2}\sin^{4}\theta \cos^{3}\left(\alpha -\varphi_{\mathrm{mid}}\right)}\right]\right\}$$

After eliminating the phase information of the along-track direction, an inverse Fourier transform of the along-track direction is performed to complete the along-track imaging; the signal is compressed as
(23)$$\begin{array}{cc} S_{6}(\tau ;x_{l};\varphi_{n}=\varphi_{\mathrm{mid}}) & =A_{5}\sin c\left\{B_{r}\left[\tau -2\frac{\Omega_{\mathrm{mid}}+x_{\mathrm{mid}}\cos\theta \cos\left(\alpha -\varphi_{\mathrm{mid}}\right)}{c}\right]\right\}\\ & \cdot \sin c\left\{\frac{2\varepsilon \sin\theta \cos^{2}\left(\alpha -\varphi_{\mathrm{mid}}\right)}{\lambda }\left(x_{l}-x_{0}\right)\right\}\\ & \cdot \exp\left\{j4\pi \frac{f_{0}}{c}R_{\mathrm{arc}}\cos\theta \cos\left(\varphi -\varphi_{\mathrm{mid}}\right)\right\}\\ & \cdot \exp\left\{-j\frac{4\pi }{\lambda }\left[\Omega_{\mathrm{mid}}+x_{\mathrm{mid}}\cos\theta \cos\left(\alpha -\varphi_{\mathrm{mid}}\right)\right]\right\} \end{array}$$

It can be obtained from the above formula that after along-track matching imaging, the actual position of the target at $\Omega_{\mathrm{mid}}$ is shifted by $x_{\mathrm{mid}}\cos\theta \cos\left(\alpha -\varphi_{\mathrm{mid}}\right)$ along the slant range, and a corresponding geometric correction is required for this. The slice data corresponding to different azimuths in the oblique range and along the track are selected in turn for two-dimensional processing and rearranging the acquired two-dimensional data according to the above steps to obtain the three-dimensional imaging of the front target. The expression is written as
(24)$$\begin{array}{cc} S_{7}(\tau ;x_{l};\alpha ) & =A_{6}\sin c\left\{B_{r}\left[\tau -\frac{2\Omega_{\mathrm{mid}}}{c}\right]\right\}\\ & \cdot \sin c\left\{\frac{2\varepsilon \sin\theta \cos^{2}\left(\alpha -\varphi_{\mathrm{mid}}\right)}{\lambda }\left(x_{l}-x_{0}\right)\right\}\\ & \cdot \sin c\left[\frac{L_{\mathrm{eff}}}{\lambda R_{m}}\left(\alpha -\varphi_{\mathrm{mid}}\right)\right] \end{array}$$
where $A_{6}$ represents the following:(25)$$\begin{array}{cc} A_{6} & =\sigma_{n}(\varphi_{n};r_{n};h_{n})\\ & \cdot \exp\left\{j4\pi \frac{f_{0}}{c}R_{\mathrm{arc}}\cos\theta \cos\left(\varphi -\varphi_{\mathrm{mid}}\right)\right\}\\ & \cdot \exp\left\{-j\frac{4\pi }{\lambda }\left[\Omega_{\mathrm{mid}}+x_{\mathrm{mid}}\cos\theta \cos\left(\alpha -\varphi_{\mathrm{mid}}\right)\right]\right\} \end{array}$$

In Equation (24), $B_{r}$ represents the range bandwidth, $B_{a}=\frac{L_{\mathrm{eff}}}{\lambda R_{m}}$ is the azimuth bandwidth, and $B_{x}=\frac{2\varepsilon \sin\theta \cos^{2}\left(\alpha -\varphi_{\mathrm{mid}}\right)}{\lambda }$ is the signal along-track bandwidth. So far, the three-dimensional imaging of the front target of the entire arc array carrier platform has been completed.

## 5. Numerical Simulation Experiments

### 5.1. Spatial Resolution Analysis

In this paper, a forward-looking AA-SAR 3D imaging model is constructed, and the corresponding imaging algorithm is proposed. In this chapter, the spatial resolution of AA-SAR is analyzed in detail when moving forward, and the simulation experiments are carried out on point targets to verify the validity of the model and algorithm. Spatial resolution, one of the important indicators used to evaluate the effectiveness of radar imaging algorithms, reflects the ability of radar to distinguish targets in different positions. The system parameters of the carrier platform are shown in Table 1, simulating the low-flying situation of the carrier platform and selecting a smaller pitch angle for the observation to reach a farther scene. To ensure that there is no ambiguity problem along the track and, at the same time, the low-altitude aircraft is safe, the aircraft is usually required to fly at a slow speed.

#### 5.1.1. Azimuth Resolution

Since the azimuth angle information corresponds to the azimuth direction information, the relationship between the target azimuth angle and the azimuth is $y_{0}=R_{x_{l}}\sin\varphi_{x_{l}}$, so we derive the azimuth resolution according to the angular resolution.

Using the geometric model, we know that the arc array SAR antenna array is arranged in turn along the angle, so the angular resolution is the array resolution. Because the instantaneous frequency in the azimuth direction is always tangent to the arc where the antenna element is located, the special antenna structure makes the direction of the instantaneous frequency change with the sampling point. The azimuth instantaneous phase of the point target can be expressed as $\frac{2\pi f_{0}R_{m}}{c}$; from this, the azimuth instantaneous frequency formula can be deduced as
(26)$$f_{\varphi }=\frac{4\pi R_{\mathrm{arc}}R_{x_{l}}\cos\theta \cos\left(\varphi -\varphi_{x_{l}}\right)}{\lambda \Omega }$$

When the observation target is very far away from the radar antenna, it is known that the azimuth half-wave width will not exceed under normal circumstances, so the value range of $\left(\varphi -\varphi_{x_{l}}\right)$ in Formula (26) is less than or equal to half of the azimuth beam width. Within this range, the azimuth instantaneous frequency is a monotonic function, and the angular resolution can be approximately expressed as
(27)$$\rho_{\varphi }=\frac{2\pi }{\max\left\{f_{\varphi }\right\}-\min\left\{f_{\varphi }\right\}}\approx \frac{\lambda \Omega }{2R_{\mathrm{arc}}R_{x_{l}}\beta }$$

Formula (27) shows that the azimuth resolution is related to the antenna radius, target slant range, and platform height and changes with the target ground distance. According to the relationship between azimuth and azimuth, to reduce the influence of azimuth changes, the scope of the detection scene is limited, and the azimuth angle is taken as a smaller value. Thus, we can approximate the relationship between the azimuth angle and azimuth as $y_{0}=R_{x_{l}}\varphi_{x_{l}}$ and obtain the azimuth information, and the azimuth resolution can be expressed as
(28)$$\rho_{y}(\varphi ;R;h)=\frac{\lambda \Omega }{2R_{\mathrm{arc}}\beta }=\frac{\lambda \sqrt{{\left(R_{x_{l}}-R_{\mathrm{arc}}\right)}^{2}+{\left(h_{\mathrm{arc}}-z_{0}\right)}^{2}}}{2R_{\mathrm{arc}}\beta }$$

To observe the effect of target height, ground distance, and point azimuth angle on the azimuth resolution, a comparative experiment was carried out. Figure 5 shows that the azimuth resolution does not depend on the azimuth angle of the target or whether the target is at the edge of the beam. The azimuth resolution of the target at the same distance is the same, but it has a certain relationship with the antenna radius and the azimuth beam width.

#### 5.1.2. Along-Track Resolution

After the above-mentioned spatial position correction and 3D imaging processing, the 3D imaging expression of the target can be obtained as
(29)$$\begin{array}{cc} S_{8}(\tau ;x_{l};\alpha ) & =A_{6}\sin c\left\{B_{r}\left[\tau -\frac{2\Omega_{\mathrm{mid}}}{c}\right]\right\}\\ & \cdot \sin c\left\{B_{x}\left(x_{l}-x_{0}\right)\right\}\\ & \cdot \sin c\left[B_{a}\left(\alpha -\varphi_{\mathrm{mid}}\right)\right] \end{array}$$

According to the along-track information contained in Equation (29), the along-track resolution can be expressed as
(30)$$\rho_{x}(\varphi ;R;h)=\frac{\lambda }{2\varepsilon \sin\theta \cos^{2}\left(\alpha -\varphi_{\mathrm{mid}}\right)}=\frac{\lambda }{2\varepsilon \frac{h_{\mathrm{arc}}-z_{0}}{\sqrt{{\left(R_{x_{l}}-R_{\mathrm{arc}}\right)}^{2}+{\left(h_{\mathrm{arc}}-z_{0}\right)}^{2}}}\cos^{2}\left(\alpha -\varphi_{\mathrm{mid}}\right)}$$

The comparison experiment of the along-track resolution is shown in Figure 6. It can be concluded that the along-track resolution will be affected by the target height, ground distance, and azimuth. In addition, it will also be affected by the forward-looking SAR beam front view angle and the elevation beam width.

#### 5.1.3. Height-Wise Resolution

Using the target slope distance $\Omega =\sqrt{{\left(R_{x_{l}}-R_{\mathrm{arc}}\right)}^{2}+{\left(h_{\mathrm{arc}}-z_{0}\right)}^{2}}$, it can be known that the height information is hidden in the slope distance information. To obtain the height resolution, we introduce the definition of the fuzzy function [28,29] and construct the fuzzy function of the target ${P_{n}}^{\prime }$, which is different from the point target $P_{n}$ in the height direction by Δh as follows:(31)$$\begin{array}{cc} S_{8}(\Delta h;x_{l}=x_{0};\alpha =\varphi_{\mathrm{mid}}) & =A_{6}\sin c\left\{\frac{2B_{r}}{c}\left[\Omega_{\mathrm{mid}}-\Omega_{\mathrm{mid}}\left(\Delta h\right)\right]\right\}\\ & =A_{6}\sin c\left\{\frac{2B_{r}}{c}\left[\Omega_{\mathrm{mid}}-\sqrt{{\left(R_{\mathrm{mid}}-R_{\mathrm{arc}}\right)}^{2}+{\left(h_{\mathrm{arc}}-z_{0}-\Delta h\right)}^{2}}\right]\right\}\\ & =A_{6}\sin c\left\{\frac{2B_{r}}{c}\frac{\left(h_{\mathrm{arc}}-z_{0}\right)\Delta h}{\sqrt{{\left(R_{\mathrm{mid}}-R_{\mathrm{arc}}\right)}^{2}+{\left(h_{\mathrm{arc}}-z_{0}\right)}^{2}}}\right\}\\ & =A_{6}\sin c\left\{\frac{2B_{r}}{c}\sin\theta \Delta h\right\} \end{array}$$

According to (31), the height resolution can be expressed as
(32)$$\rho_{z}(\varphi ;R;h)=\frac{c}{2B_{r}\sin\theta }=\frac{c}{2B_{r}\frac{\left(h_{\mathrm{arc}}-z_{0}\right)}{\sqrt{{\left(R_{\mathrm{mid}}-R_{\mathrm{arc}}\right)}^{2}+{\left(h_{\mathrm{arc}}-z_{0}\right)}^{2}}}}$$

In (32), the height resolution of the forward-looking AA-SAR imaging area depends on the slant-range bandwidth of the radar transmit signal and the imaging geometry of the radar; its value is equal to the slant-range resolution of a typical SAR multiplied by the reciprocal of the sine value of the viewing angle of a beam front. This shows that the height resolution mainly comes from the effective bandwidth or effective time width of the radar transmit signal in the height direction. Figure 7 is a comparison experiment of different target height resolutions. It can be seen that the height resolution is affected by the target slant range and target height in the imaging geometry, but the target azimuth does not affect the height resolution.

### 5.2. Point Target Imaging Analysis

To verify the effectiveness of the model and algorithm, we conducted simulation experiments on point targets. The simulation results were analyzed as follows.

Figure 8a is the distribution map of the real position of the target in the imaging scene, and Figure 8b is the imaging result of the target in the observation scene.

Point targets P5 and P9 were selected for measurement and analysis to further verify the effectiveness of the proposed imaging method. Figure 9 shows the single-point 3D imaging of point targets P5 and P9, which shows that the imaging point target has good focusing performance.

Figure 10 shows the cross-sectional view and response function of point target P5 in three dimensions, and Figure 11 shows the cross-sectional view and response function of point target P9 in three dimensions, which shows that the focusing effect of the two-point targets in the along-track, azimuth, and height directions is good.

The two-dimensional slice diagram of the point target in Figure 10 shows that three-dimensional imaging can be well realized because there is no azimuth difference at this position. The point targets in Figure 11 have an azimuth difference, so there is a partial mismatch in the two-dimensional slice image. In addition, the three-dimensional response functions of point targets are all sinc functions, which shows that the resolution of the airborne forward AA-SAR three-dimensional mode meets the imaging requirements, which further verifies the effectiveness of the arc array SAR three-dimensional imaging algorithm based on keystone transform studied in this paper.

## 6. Conclusions

Arc array 3D imaging will have greater advantages in the future, with a wider market and a wider imaging field of view. To quickly obtain omnidirectional imaging and overcome the shortcomings of traditional array SAR 3D imaging, this paper proposes a 3D scene imaging mode suitable for arc arrays in a low-altitude ground-based reconnaissance mode. Based on the premise of not increasing the complexity of the original system, the AASAR platform is used to move forward to obtain the second synthetic aperture, and a corresponding imaging algorithm is proposed for the above modes. Because of the special geometric configuration of AASAR, the external loader platform moves forward at a constant speed, resulting in complex antenna motion trajectories, which brings difficulties to subsequent imaging. This paper proposes a corresponding solution, which simplifies the AASAR forward mode into two steps: First, the target distance equation is simplified, the target azimuth angle is discussed, and the influence of the forward motion of the platform on the along-track position is analyzed to further realize the two-dimensional focusing of the target slant range–azimuth direction. Second, a new azimuth angle variable is redefined in the slant-range on-track imaging, and a 3D imaging algorithm based on keystone transformation is proposed to eliminate the coupling term generated by the array angle and slant range time to realize the 3D imaging of the target. In addition, the spatial resolution of arc array SAR is analyzed in detail, focusing on the relationship between the target azimuth and azimuth. Finally, the algorithm’s effectiveness is verified through numerical simulation experiments involving point targets.

## Figures and Tables

**Figure 1 sensors-23-02674-f001:**
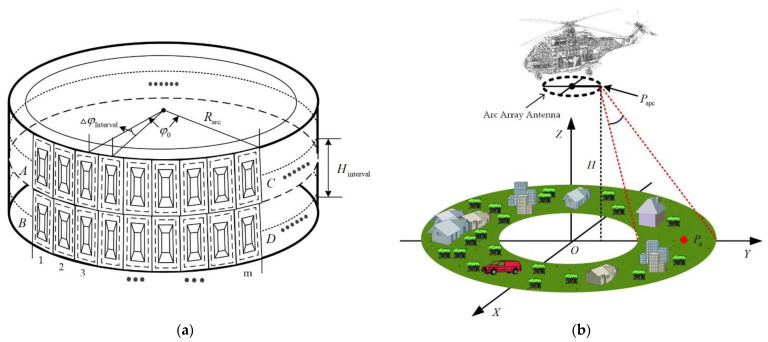
Antenna array and imaging geometry. (**a**) Arc array antenna distribution diagram; (**b**) geometric scenarios for arc array SAR imaging.

**Figure 2 sensors-23-02674-f002:**
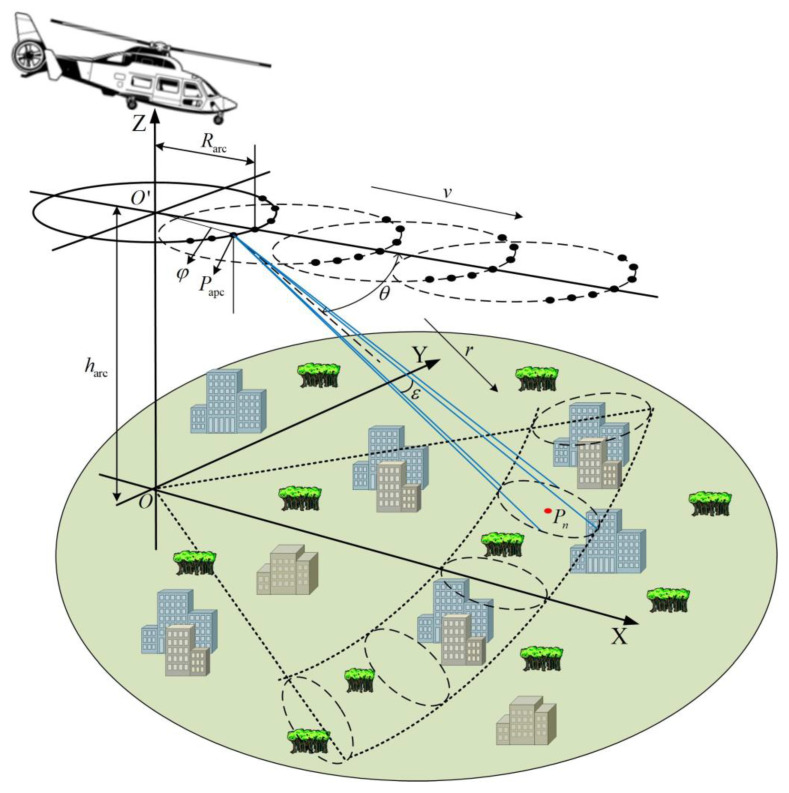
AASAR observation scene model in forward mode.

**Figure 3 sensors-23-02674-f003:**
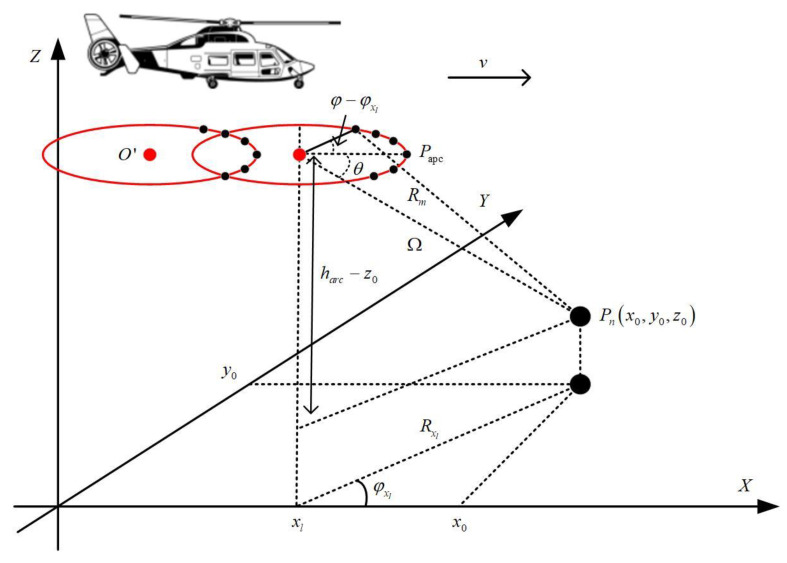
Azimuth section of AASAR in forward mode.

**Figure 4 sensors-23-02674-f004:**
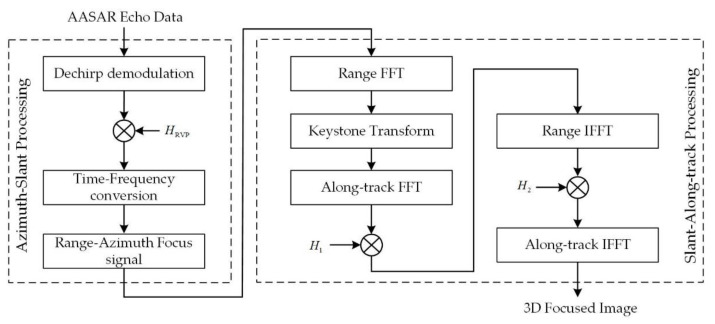
Arc array SAR flow diagram.

**Figure 5 sensors-23-02674-f005:**
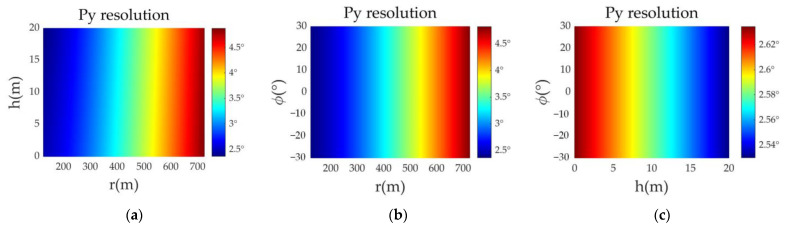
Azimuth resolution of targets at different positions. (**a**) The target azimuth angle is 30°; (**b**) the target height is 10 m; (**c**) the target distance is 200 m.

**Figure 6 sensors-23-02674-f006:**
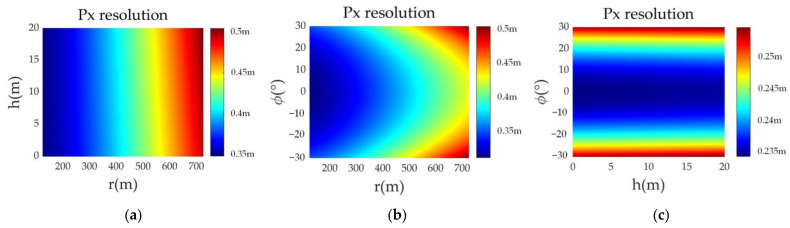
Along-track resolution of targets at different positions. (**a**) The target azimuth angle is 30°; (**b**) the target height is 10 m; (**c**) the target distance is 200 m.

**Figure 7 sensors-23-02674-f007:**
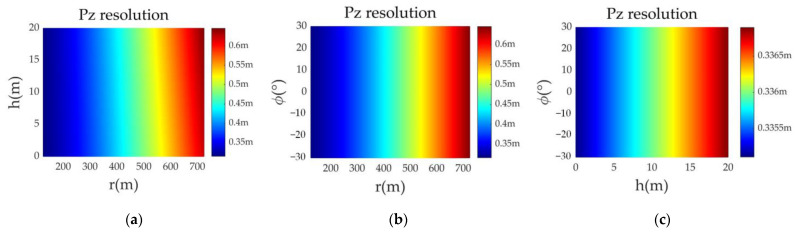
Height-wise resolution of targets at different positions. (**a**) the target azimuth angle is 30°; (**b**) the target height is 10 m; (**c**) the target distance is 200 m.

**Figure 8 sensors-23-02674-f008:**
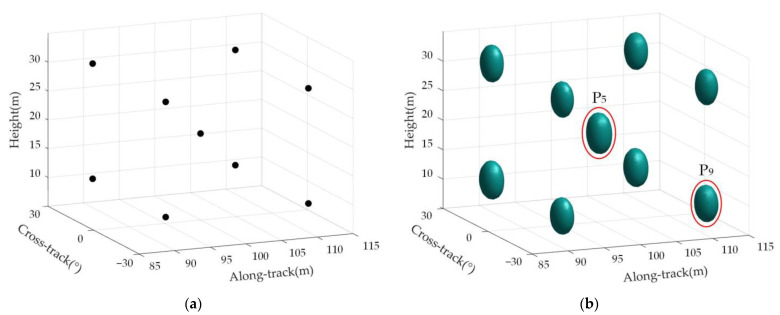
Multi-point target position and the imaging result of targets. (**a**) Multi-point target distribution location map; (**b**) 3D imaging results of multi-point targets.

**Figure 9 sensors-23-02674-f009:**
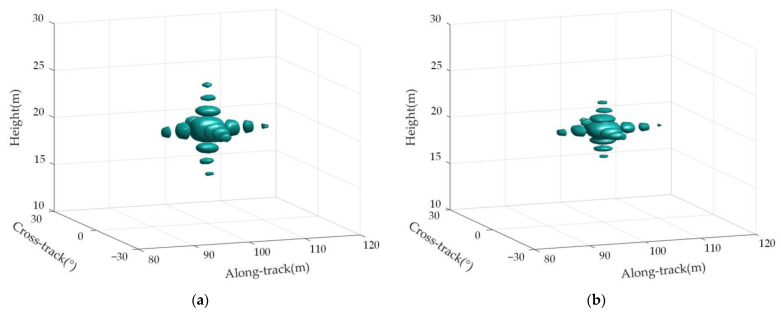
A 3D image of single-point targets. (**a**) Point P5 3D image; (**b**) point P9 3D image.

**Figure 10 sensors-23-02674-f010:**
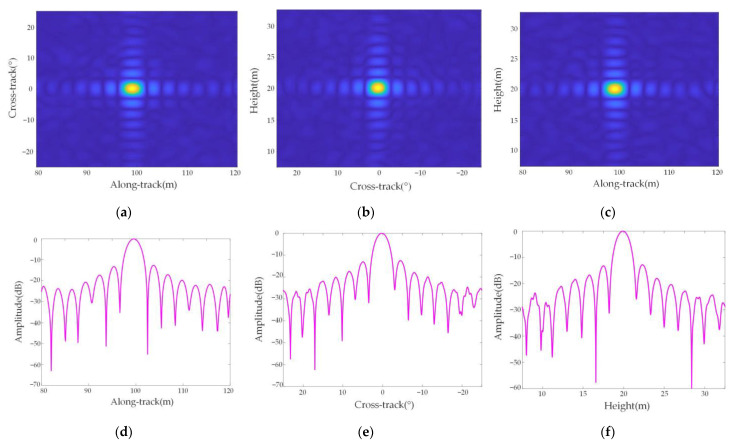
Point P5 3D sectional view and point target quality analysis. (**a**) Along-track-cross-track sectional view; (**b**) cross-track-height direction section view; (**c**) Along-track-height direction section view; (**d**) Along-track PSLR; (**e**) Cross-track PSLR; (**f**) Height PSLR.

**Figure 11 sensors-23-02674-f011:**
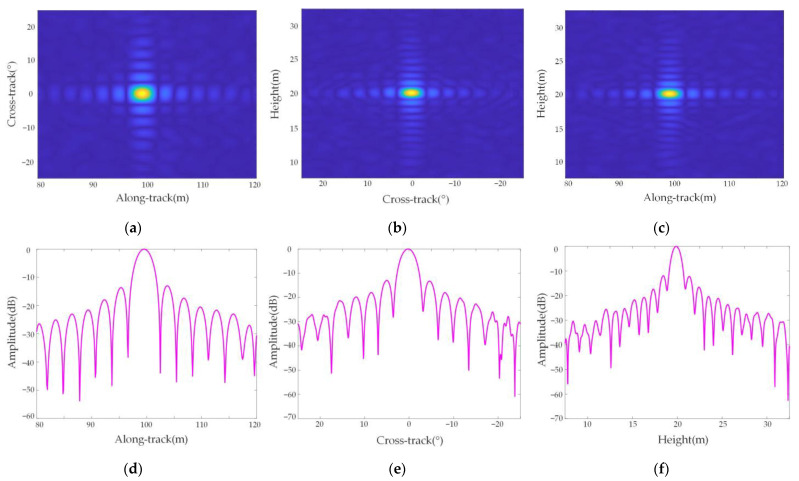
Point P9 3D sectional view and point target quality analysis. (**a**) Along-track-cross-track sectional view; (**b**) cross track–height direction section view; (**c**) along track–height direction section view; (**d**) along-track PSLR; (**e**) cross-track PSLR; (**f**) height PSLR.

**Table 1 sensors-23-02674-t001:** Parameters of arc array SAR system.

Symbol	Definition	Value
$f_{0}$	Carrier Frequency	35.5 GHz
$B_{r}$	Signal Bandwidth	800 MHz
$R_{\mathrm{arc}}$	Arc Array Radius	0.6 m
β	Array Beamwidth (−3 dB)	56°
ε	Elevation Beamwidth (−3 dB)	30°
$h_{\mathrm{arc}}$	Carrier Platform Height	500 m
θ	Platform Pitch Angle	60°
v	Platform Forward Speed	40 m/s

## Data Availability

Not applicable.

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
