# Peer review of "A Modified Keystone-Based Forward-Looking Arc Array Synthetic Aperture Radar 3D Imaging Method"

_sensors, 2023, doi:10.3390/s23052674_

Round 1
Reviewer 1 Report
In this paper, a forward-looking arc array synthetic aperture radar 3D imaging algorithm based on the Keystone transform is proposed. The topic is very interesting. The proposed method is innovative. However, in order to improve the manuscript, some small issues need to be considered.
1. The abstract is too long. It is recommended to simplify the abstract.
2. In section 1, it is suggested to discuss about the innovation of the proposed method.
3. In section 2.1, the description of the imaging mode is not clear.
4. In Section 5.2, the simulation results need to be described in more detail to show the effectiveness of the proposed method.
5. The format of the equations need to be rechecked, such as equation 29 and 31.
6. Conclusion needs to be improved. It is suggested to emphasize the main contributions of the paper.
Reviewer 2 Report
This paper proposes a three-dimensional target reconstruction using a specifically designed synthetic aperture radar (SAR). The proposal sounds interesting and simulation results seem fine. On the other hand, there are several points to be clarified and improved.
1. Frist of all, the definition of “improve” is unclear. The authors wrote “an improved 3D imaging algorithm” or “an improved keystone-based processing algorithm” while there seems no clear definition what was improved or how it was improved. Experimental results must contain a clear comparison to show the improvement. Acquiring a rigorous solution is not an improvement of the algorithm itself.
2. The authors cited multiple preceding papers related to arc array SAR (AA-SAR) while their differences or contributions are unclear. If the experimental setups in this paper is the same of the previous one, cite and describe it properly.
3. Experimental setups are unclear in Section 5. The parameters firstly appear in Section 5.2 while it is apparently applied in 5.1. Please rewrite the descriptions and provide all parameters so that readers can follow the experiments.
4. As the authors assume to mount 35GHz center frequency SAR on a helicopter, orbit stability is critical for the processing. Please discuss in such a point of view.
5. In Fig. 2 the authors draw an airplane while in the body text in L. 142, it is helicopter.
6. In L 147, the authors wrote it is polar coordinates while the equations are written in cylindrical coordinates.
Round 2
Reviewer 2 Report
The manuscript seems fine to be published.